# Why 'elevating country voice' is not decolonizing global health: A frame analysis of in-depth interviews

**Michael Kunnuji**[1]*, **Yusra Ribhi Shawar**[2,3]*, **Rachel Neill**[2], **Malvikha Manoj**[2], **Jeremy Shiffman**[2,3]

**1** Department of Sociology, University of Lagos, Akoka, Lagos, Nigeria, **2** Department of International Health, Johns Hopkins University Bloomberg School of Public Health, Baltimore, Maryland, United States of America, **3** Johns Hopkins University Paul H. Nitze School of Advanced International Studies, Johns Hopkins University, Washington, DC, United States of America

* michaelkunnuji@gmail.com (MK); yusra.shawar@jhu.edu (YRS)

**Data Availability Statement:** The study used primary qualitative in-depth interviews. The authors are not able to share the transcripts publicly given

## Abstract

Recent calls for global health decolonization suggest that addressing the problems of global health may require more than 'elevating country voice'. We employed a frame analysis of the diagnostic, prognostic, and motivational framings of both discourses and analyzed the implications of convergence or divergence of these frames for global health practice and scholarship. We used two major sources of data–a review of literature and in-depth interviews with actors in global health practice and shapers of discourse around elevating country voice and decolonizing global health. Using NVivo 12, a deductive analysis approach was applied to the literature and interview transcripts using diagnostic, prognostic and motivational framings as themes. We found that calls for elevating country voice consider suppressed low- and middle-income country (LMIC) voice in global health agenda-setting and lack of country ownership of health initiatives as major problems; advancing better LMIC representation in decision making positions, and local ownership of development initiatives as solutions. The rationale for action is greater aid impact. In contrast, calls for decolonizing global health characterize colonialityas the problem. Its prognostic framing, though still in a formative stage, includes greater acceptance of diversity in approaches to knowledge creation and health systems, and a structural transformation of global health governance. Its motivational framing is justice. Conceptually and in terms of possible outcomes, the frames underlying these discourses differ. Actors' origin and nature of involvement with global health work are markers of the frames they align with. In response to calls for country voice elevation, global health institutions working in LMICs may prioritize country representation in rooms near or where power resides, but this falls short of expectations of decolonizing global health advocates. Whether governments, organizations, and communities will sufficiently invest in public health to achieve decolonization remains unknown and will determine the future of the call for decolonization and global health practice at large.

ethical obligation, as determined by Johns Hopkins IRB, where we received exemption on the condition that data collected not be shared beyond study authors. The authors made relevant excerpts of the transcripts available in the paper, however.

**Funding:** This study was made possible by the generous support of the American people through the US Agency for International Development (USAID) under the terms of the Cooperative Agreement #7200AA20CA00002, led by Jhpiego and partners. The contents are the responsibility of the authors and do not necessarily reflect the views of USAID or the United States Government. The funders had no role in the study design, data collection and analysis, decision to publish, or preparation of the manuscript.

**Competing interests:** The authors have read the journal's policy and have the following competing interest: RN was an independent consultant for MOMENTUM on a different topic during a portion of this study. The authors declare that they have no other competing interests. This does not alter adherence to PLOS ONE policies on sharing data and materials.

## Introduction

Global health is typically framed with a normative objective of achieving equity in access to health among people all over the world, a feature that sets it apart from tropical medicine or international health [1]. Despite its normative objective, there are concerns by some actors that global health actually perpetuates the power imbalances and resulting ills that were historically manifest between colonized and colonizer states–that there is inadequate representation of low-and-middle income country (LMIC) actors' voices in agenda setting, and/or insufficient LMIC ownership of health initiatives. This has resulted in recent calls by some actors to 'decolonize global health' [2–7], and others to 'elevate country voice' [8, 9]. We investigate the extent to which attempts to 'elevate country voice' aligns with the expectations of those calling for the 'decolonization of global health' and discuss the implications of convergence or divergence of the frames that underly these discourses for global health practice and scholarship. In this study, we define country voice as national priorities as defined by local actors in LMICs. Elevating country voice thus implies local leadership of development initiatives.

Both calls to 'decolonize global health' and 'elevate country voice' seek to address power asymmetry in global health, and stem in part from a concern that LMIC representation in global health governance is inadequate, unequitable, and requires change. Yet, the problems, solutions, and progress markers are framed differently, with different implications for acceptance among global health funders and decision makers. It is therefore important to examine the historical contexts and origins of the two concepts, as well as the main actors involved in advancing them, as this is likely to influence how the concepts are interpreted and in what circles they are accepted or resisted. For example, advocates of global health decolonization are likely to push back against calls to elevate country voice if it is perceived to retain ideas central to the existing power dynamics and suggest that current power brokers must be the ones to elevate local country voice. Likewise, those advancing the need to elevate country voice are likely to oppose calls to decolonize global health if it is perceived to fundamentally denounce and call into question well-intentioned efforts of high-income country (HIC) actors working in LMICs.

This analysis aims to help global health researchers, practitioners and policymakers understand how calls for 'decolonizing global health' and 'elevating country voice' shape global health research and practice broadly, and to understand the changes each of the calls may bring about in global health research and practice in the future.

### Frame analysis as an explanatory tool

The article is built on frame analysis which has its origin in the works of Erving Goffman [10]. Goffman explained that frames are definitions of situations that are built up in line with the principles that govern events and people's subjective involvement in such events [10]. People, therefore, make sense of issues differently based on how the issues are framed, that is, presented to them and their actions are guided by frames [11]. Frames highlight what is relevant; link the various highlighted elements in a single story so that one set of meanings is conveyed; and shape the way specific elements are seen in relation to other things [12]. By implication, the same event may be given different meanings when viewed through different frames [13].

Between the broad perspectives of actors calling for elevated country voice and those calling for decolonization of global health, there is agreement on the need for change in global health practice. If these calls to action are to result in change, they need to align and resonate with the existing worldviews of those who will be involved in the change process. The resultant frame alignment is necessary for mobilizing actors for change [14, 15]. Snow and Benford [15] further identified three core framing tasks–diagnostic framing, which involves identification of a

problem and the attribution of blame or causality; prognostic framing, which suggests solutions and identifies strategies, tactics and targets; and motivational framing, which is the elaboration of a call to arms or rationale for action. We contrast the historical contexts of elevating country voice and decolonizing global health and their underlying diagnostic, prognostic and motivational framings, drawing from how both concepts are understood by key actors in a U. S. Agency for International Development (USAID) suite of awards aimed at improving maternal, newborn and child health services and voluntary family planning in selected LMICs. Our focus on actors affiliated with these USAID-funded set of awards offers an illustrative case of larger global health dynamics given the agency's major role in global health practice, especially in the areas of maternal and child health, HIV/AIDS, and other infectious diseases across many LMICs.

## Methods

### Data sources

We used two major sources of data: in-depth interviews with key actors within the USAID MOMENTUM suite (established to improve maternal, newborn, and child health services and voluntary family planning) and closely adjacent partners; and a review of literature. The USAID MOMENTUM is a suite of six maternal, newborn, child health, voluntary family planning, and reproductive health projects in 28 LMICs. The MOMENTUM projects adopt an approach that increases the commitment of host countries and capacity of partner institutions and local organizations to deliver services in selected countries. In principle, the MOMENTUM approach aligns with the quest for improved LMIC representation in global health governance, which underlies calls for both elevating country voice and decolonizing global health. In addition, the suite's wide geographical spread across several LMICs makes it an interesting case study that those working in similar settings may draw lessons from. These explain our choice of the MOMENTUM suite for this analysis.

We conducted semi-structured in-depth interviews with 27 actors (seven in Africa, three in Asia, one in Australia, one in Europe, and 15 in North America) within the MOMENTUM suite and global health thought leaders, especially those who have expressly advocated for elevating country voice or decolonizing global health in writing (Table 1). These interviewees included USAID staff, staff of international non-governmental organizations (NGOs) at the headquarters and country level, academics, and government officials in Ghana, Indonesia, India, Pakistan, South Africa, and Tanzania, which are MOMENTUM focal countries. We conducted the interviews via Zoom between March and June, 2021. The interviews touched on a wide range of topics, including health systems resilience, self-reliance, and elevating country voice and/or decolonizing global health. We asked questions on: how interest in decolonizing global health/elevating country voice began in global health circles; concepts used before the introduction of decolonizing global health/elevating country voice; how actors understood the decolonizing global health/elevating country voice discourse, how decolonized global health/elevated country voice can be measured; and how the idea of decolonization of global health/elevated country voice is shaping global health practice around the world. The study protocol underwent ethics review and received exemption by the Institutional Review Board of Johns Hopkins University in Baltimore, MD, USA, as it focused on public policy, did not involve any vulnerable group, and was deemed to pose minimal risk to informants.

While this is **not** a systematic review, we conducted a literature search to purposively select published articles that used the key concepts of decolonization of global health/decolonizing global health, elevating/increasing country voice, resilience, and self-reliance, since 2000. The search was done on October 6, 2020 but articles with the same thematic focus published after

**Table 1. Location, origin and organizational roles of interviewees.**

| Interview Number | Geographical Location—Country | Geographical Location—Content | Origin | Organization Role/Affiliation |
|---|---|---|---|---|
| I1 | LMIC | Africa | HIC | Implementing Partner |
| I2 | LMIC | Africa | HIC | Implementing Partner |
| I3 | LMIC | Asia | HIC | Implementing Partner |
| I4 | HIC | North America | HIC | Implementing Partner |
| I5 | HIC | North America | LMIC | Implementing Partner |
| I6 | HIC | North America | LMIC | Implementing Partner |
| I7 | HIC | North America | LMIC | Intergovernmental Organization |
| I8 | LMIC | Asia | LMIC | Implementing Partner |
| I9 | HIC | North America | HIC | Implementing Partner |
| I10 | HIC | North America | HIC | Donor |
| I11 | HIC | North America | HIC | Intergovernmental Organization |
| I12 | HIC | North America | HIC | Implementing Partner |
| I13 | HIC | North America | HIC | Implementing Partner |
| I14 | HIC | North America | HIC | Donor |
| I15 | LMIC | Africa | LMIC | Academic |
| I16 | LMIC | Africa | LMIC | Government/Academic |
| I17 | HIC | North America | LMIC | Implementing Partner |
| I18 | LMIC | Africa | LMIC | Government/Academic |
| I19 | LMIC | Africa | LMIC | Government |
| I20 | HIC | North America | HIC | NGO |
| I21 | HIC | Europe | HIC | Intergovernmental Organization |
| I22 | HIC | North America | HIC | Donor |
| I23 | LMIC | Asia | LMIC | Academic |
| I24 | HIC | North America | HIC | Academic |
| I25 | LMIC | Africa | LMIC | Academic |
| I26 | HIC | Australia | LMIC | Academic |
| I27 | HIC | North America | LMIC | Donor |

this date were later added. From the initial review, we retrieved 512 articles and uploaded them into RefWorks for review. One of the authors reviewed the titles and abstracts of the decolonization articles, and articles were included for analysis if they were explicitly about decolonizing global health or elevating/increasing country voice within global health. The exclusion criteria used include mentioning colonialism without engaging in anti-colonialist thought; not addressing issues within the health sector, applying a decolonizing approach in high-income countries only; focusing on power relations between medical volunteers/students and not addressing health systems programs/policies/technical assistance; and not addressing the topic of decolonization of global health/elevating country voice. A total of 79 articles met the inclusion criteria. Additional peer-reviewed literature was also identified using snowballing approaches from the reference list of included studies, including relevant publications prior to 2000. We did a second search for relevant gray literature using the Google search engine. For this, the search concepts and terms used in the original review were used. Gray literature sourced from USAID and its partners was prioritized due to the aims of the review.

## Data analysis

A deductive analysis approach was applied using the frame theory, with diagnostic framing, prognostic framing and motivational framing as key thematic areas. Concept origin was

also included as a theme. We analyzed both the literature and interview transcripts using Nvivo 12.

## Results

We first present the history of the frames. Then, we analyze and compare the diagnostic, prognostic and motivational frames of both elevating country voice and decolonizing global health.

### Historical contexts of the frames

At about the turn of the century, major actors in international development initiatives had come to terms with the need for improving the mode of operation in development work. This was motivated more out of the need to make aid effective and yield greater impact, rather than out of the need to address any perceived ills in international relations (I15) [8, 16]. In 2003, the Organization for Economic Cooperation and Development (OECD) facilitated a meeting of multilateral and bilateral development institutions and aid recipient countries to harmonize their operational policies, procedures and practices to ensure the achievement of the Millennium Development Goals [16]. The forum acknowledged concerns by recipient countries that 'donors' practices do not always fit well with national development priorities and systems' [16]. In 2005, at a follow up forum, donor and recipient countries committed to five principles, notable among which are country ownership and leadership of their own development work, and alignment of aid to local priorities [8]. The Paris Declaration, as the output of the 2005 forum came to be known, then became a reference point in attempts to elevate country voice in international development work involving donor and recipient countries, including global health initiatives. A third high-level forum in 2008, which built on the Paris Declaration, further focused attention on country ownership, building more effective and inclusive partnerships, and achieving development results and openly accounting for them [17].

This happened partly because in the early 2000s, there were lots of "questions around why global health interventions fail" (I15) and also because capacity to own and lead initiatives were increasingly recognized in LMICs, the result being an increased push by local actors for their voices to be heard (I8). The principles of the Paris Declaration have continued to guide major organizations' development work. In 2010, for instance, USAID undertook a major reform agenda known as 'USAID Forward' with three main areas of focus, one of which is to 'promote sustainable development through high-impact partnerships and local solutions'. With this reform, USAID missions use, strengthen and partner with local actors, including governments, civil society organizations, and local private sector actors [18]. The push for the realization of the goals of the Paris Declaration is reflected in the elevating country voice frame.

In contrast, the decolonizing global health discourse rests on a historical and conflict perspective which emphasizes the domination of a social group by another. The colonization of the rest of the world by European countries led to the racial hierarchization of humanity (i.e., the creation of a system of white supremacy), exploitation, domination, and control of all facets of life of the colonized world by the colonialists [19, 20]. After the independence of colonized states, international relations retained its coloniality (I15) [21] which manifests in present day racial, political-economic, social, epistemological, linguistic, and gendered hierarchical orders. Health initiatives involving multiple countries of the world betray these unequal power relations [3–7]. For instance, in the colonial era and afterwards, people conceptualized health problems and the structure of health services in ways shaped by the economic, social and political requirements of colonialists, instead of the health needs or preferences and

tradition of the people [22, 23]. The colonial medical system was exclusively biomedicine [24, 25] which has been described as rooted in racialized Eurocentrism [26].

Tropical medicine and international health, both precursors of global health, are viewed by many in the decolonizing global health movement as producers of the processes of othering and dehumanization of the colonized in health programming. Global health, in spite of its normative objective of addressing global health inequities and inequalities, keeps this tradition of power asymmetry [3, 6, 27]. Global health policy and decision-making positions are reserved for HIC actors [28–31], perpetuating a culture of white supremacy [32–35]. Low public investment in health initiatives in LMICs, which privileges donors in global health agenda setting [36], is also attributed to the exploitation of colonialism [37, 38]. These ideas of power asymmetry in global health governance and agenda setting, medical and public/global health training, and exploitative global health practice underlie the call to decolonize global health. These ideas featured first in public health contexts in countries with settler colonialism like Bolivia, Jamaica, Guatemala, and South Africa [26, 39, 40].

Recent calls to decolonize global health are a part of broader movements to decolonize wider aspects of life. It appears to have been impacted most by the 2015 Rhodes Must Fall (RMF) movement, a students' protest which demanded a more inclusive educational system and a decolonized curriculum among other things in post-Apartheid South Africa (I25) [41]. The success of the RMF movement inspired the RMF protests in Oxford [41], and the creation of the Duke Decolonizing Global Health Working group by a group of students 'to have more in-depth conversations about global health and its roots in oppressive systems' in 2018 [42]. Between 2019 and 2020, several student conferences were organized on decolonizing global health. At about the same period, major global health journals, including *BMJ Global Health* and *Lancet*, began to give attention to the topic and supported the movement's proliferation [4, 27, 37]. The COVID-19 pandemic further aided the movement given its demystifying effect of watching the health systems of HICs struggle to contain the pandemic; the scramble for medical supplies which would have been previously associated with LMICs only [6, 43]; and the infamous French doctors' conversation on the prospect of a BCG for COVID-19 treatment trial in Africa, further affirming the view that global health practices are exploitative, racist and unethical when LMICs are involved [36, 37, 44–46]. Box 1 is a summary of key developments in the histories of both concepts.

## Diagnostic framing–problem identification

We present here the frames underlying elevating country voice and decolonizing global health, and what they identify as the problem in global health and its cause.

**Elevating country voice.** The diagnostic frame of elevating country voice presents ineffective aid as the problem of global health and attributes this problem to lack of inclusiveness in global health governance and decision-making roles, suppressed LMIC voice and amplified voice of HIC actors in global health agenda setting, and lack of ownership of global health initiatives by LMICs (I1, I23, I25) [29, 47]. A respondent argued that the majority of members of the working groups of global health initiatives are individuals from Europe and North America, representing donor foundations and research organizations (I14). Referring to a personal experience with one such working group, the respondent argued:

> *"We had an initial meeting and I realized that there wasn't a single person in that meeting from any of our partnering countries and, like this is absolutely ridiculous; like why are we all coming together trying to decide what we should be focusing on and why are we not listening to our colleagues who are on the front lines and dealing with post-partum haemorrhage on a daily basis?"* (I14).

### Box 1: Historical contexts of elevating country voice and decolonizing global health concepts

| (Approx.) date | Elevating country voice | Decolonizing global health |
|---|---|---|
| ~ 1415–1980 | | European colonial rule |
| 2003 | OECD meeting of development institutions and aid recipient countries | |
| 2005 | Paris Declaration | |
| 2008 | Third OECD high-level forum | |
| 2010 | USAID Forward | |
| 2015 | | Rhodes Must Fall movement at the University of Cape Town, South Africa |
| | | Oxford Rhodes Must Fall protests |
| 2018 | | Creation of Duke Decolonizing Global Health Working Group |
| 2019–2020 | | Decolonizing Global health conferences<br>• Harvard TH Chan School of Public Health students, US– 2019<br>• School of Global Health at the University of Copenhagen, Denmark– 2019<br>• Duke Decolonizing Global Health Working Group, Duke University, US– 2020<br>• The Decolonize Global Health Working Group at the University of Edinburgh, UK– 2020<br>• Johns Hopkins University– 2020<br>• Karolinska Institutet, Stockholm, Sweden—2020 |

Another respondent further stressed how aid tilts the scale in favor of HIC actors and suppresses LMIC voice in global health:

*"I think there's been this dynamic always of he who has the dollar holds the power and so this feeling of, you know, donors having their agenda or governments having their agenda, that they'd then go in and try to get the receiving or the recipient country to go along with, as opposed to starting with where those priorities are"* (I5).

Another respondent reflected on how those holding the resources shape global health given power differentials:

*"It's very hard because the development partners are the ones who are funding things. . .. So, they actually ultimately control the power. So, they have the power in the room. The countries do not have the power in the room"* (I22).

Funders prioritize immediate results to justify investment in global health initiatives above local priorities and long-term outcomes. They have complex application submission and reporting systems that local actors are not familiar with. The implication is that only large HIC organizations that are experienced in working in the terrain, and which have previously received and managed grants by these international donors, will continue to get funded. Local organizations can hardly compete with them (I14; I22). Even when funders wish to work with

local organizations, the complex application and reporting systems and desire to get results compel them to stick with established HIC organizations while working in LMICs (I14).

As a result, the aid provided is not as impactful in LMICs—a major problem raised in the 'elevating country voice' frame. According to a respondent:

"... *the impact of aid has been minimal on countries of the South because of the idea of controlling the aid and managing the aid, not just the funding, but its implementation, its direction, the conceptual elements, it's all being done by the powerful countries of the North...*" (I23).

**Decolonizing global health.**   The decolonizing global health frame, by contrast, presents racialized hierarchization of humanity and health systems, and exploitative neoliberalism as the problems of global health. Consequently, this frame attributes blame to systemic racism (I1, I10, I15) and neoliberalism and its purveyors (I1, I15). It suggests that the problem of suppressed LMIC voice in global health agenda-setting results from coloniality. One respondent reflected:

"... *the way that we approach international development is still Europe and the US donating money or funding projects in Africa and there's still that hierarchy ...*" (I20).

In line with the diagnostic framing of racialized hierarchization of humanity and health systems, Affun-Adegbulu & Adegbulu [27] point to the lack of ontological and epistemic pluralisms in the conceptualization of humanity in global health circles as the problem. Global health initiatives run on the assumption of a superior Western medical knowledge and health practice, and this fails to give room for the integration of alternative knowledges and healing systems into health systems of countries. Where this integration appears to happen, it is one-sided, and the conditions and terms of integration are determined by biomedical professionals [48]. The result is a health system that is unjust, non-inclusive, lacking in diversity, and unable to offer equal access to health [4].

Racialized hierarchization of humanity manifests in a notion of supremacy and a mentality of saviorism in HIC actors involved in global health working in LMICs (I14). According to some respondents, many HIC actors side-track and delegitimize governments in LMICs (I21) and work with the mentality that *"Well they're not going to do it for themselves and so we've got to come in and do it for them"* (I14). A couple respondents reflected on the underlying racism that underpins global health practice:

"*I think in the way we think and practice global health, you know, the idea that knowledge flows from the North to the South, you know, still persists. The idea that, you know, there's this kind of burden, the white man's burden, right, to save the black man or to save the rest of the world. That still persists, which again it still has tinges of racism in it*" (I15).

"*We [HIC actors] were not accepting the expertise and the leadership in the countries where we were working. We have come in again as helpful colonists...*" (I10).

Akugizibwe [36] employed the frame of exploitative neoliberalism while arguing that global health operates a system that treats LMICs as subjects of HICs' philanthropy, but in reality, this system exists to serve the economic interest of HICs. Global health is therefore an exploitative tool which needs to be examined through the lens of the global political economy. Stretching this argument, Akugizibwe [36] argued that donors provided more than a fifth of

health spending in 20 sub-Saharan African countries, and over 40 percent of health spending in nine countries. While this is often framed as a one-way flow of charity, it is in reality an indication of a complex power dynamic that yields benefits for the donors by improving their relations with the recipient countries, giving them control over global health agenda and establishing an incentive for those LMICs to align their policies with the interests of donor [28, 36]. This philanthropy also safeguards HIC pharmaceutical industry profits [2]. The problem of exploitative neoliberalism is attributed to lack of government investment in health systems in LMICs [28]. Key international global health players such as the World Bank give aid/loans that come with conditions, including the inclusion or exclusion of specific programs (I7). In this way, the priorities of LMICs are determined by HIC players.

A respondent supported the view that economic interests often override positive health outcomes in global health practice:

*"For at least ten years, if not more, I've heard low- and middle-income countries saying we want the ability to produce our own vaccines in our countries or at least in our regions; and XXX was always like this is impossible, it cannot be done. . ."* (I27).

The words used to describe different parties in global health initiatives are also suggestive of absence of equity; a sentiment highlighted by this respondent:

*"And if you look at the amount of money that comes from donors to low- and middle-income countries, sometimes it's a drop in the ocean compared to the national budget. It's not always, but in some cases, it is, right. So why are we calling them donors and why are we saying recipient countries rather than implementing countries*?" (I27).

The respondent explains further:

*"So even the language that we use or, you know, even at the XX Foundation I think it's even worse, we call them markets. You know, they're not even countries anymore. They're markets*! *. . . and the incentive is to get results from your grantees, you have a lower return on investments sometimes with a local partner than with an international partner that understands the dynamics of the XX Foundation"* (I27).

Since LMICs are seen as markets, respondents also argued that HIC organizations compete for returns on investment and prioritize justifying to their government how they have used 'taxpayers' money', giving less attention to the involvement of local LMIC organizations or what recipient countries consider priority in the implementation of health initiatives (I7).

Decolonizing global health also considers the way LMICs are represented on the boards and working groups of major global health organizations and initiatives. Some respondents argued that people are often selected to represent LMICs, simply because they are friends with the HIC actors in charge (I17). The same set of people are invited repeatedly as 'country voice' (I11). Sometimes the representatives lack the technical competence to elevate country voice (I16); and they are often an obstruction to elevating country voices because if true representation happens, the power of these specific representatives is diminished as prioritizing the interests of LMICs often does not serve their personal interests (I15, I17). When representatives have conflict of interests, it is difficult to know the priorities of the LMICs they represent (I10, I15, I17). Sometimes, senior government officials are bought off with development assistance money (I21), and the hierarchy in government bureaucracy unfortunately gives little room for junior officers with contrary views to present them. So, the decolonizing global health frame

queries the logic behind equating tokenism or the invitation of a few representatives to an agenda-setting meeting with elevating the voice of the entire country (I10, I22). A respondent asked rhetorically:

'*Okay, we threw in a country voice, we had this one tokenistic person show up and we can now check the box*!?*'* (I10).

Further complicating representation is the cost and logistics of participating in meetings that hold in Geneva or Washington, which naturally skew the table against LMICs (I27).

Within the decolonizing global health frame, widespread attitudes and beliefs that support coloniality in global health practice among key LMIC actors are seen as a problem which they call 'colonialism of the mind'. One respondent said:

"*That's really the worst type of colonialism, [the] colonialism of the mind, you know, because it affects the way you think about yourself, it affects the way you think about your abilities, but also affects the way you think about the white man, you know, or woman, and so there's, I think, that mindset still exists in the South where, you know, some still expect ideas to come from the North*" (I15).

Finally, the global health decolonization diagnostic frame situates global health within a larger political and economic structure of power asymmetry beyond global health practice and policies themselves. It posits that global health purports to operate in an apolitical vacuum, but in reality, it operates within an international political and economic context that perpetuates inequality among nations, and poverty in LMICs, the result being poor health outcomes and the need for aid in LMICs [21, 49].

## Prognostic framing—Solutions and strategies

The prognostic framing, that is, suggested solutions, strategies and tactics for addressing problems and progress markers of the elevating country voice and decolonizing global health frames, are presented in this section.

**Elevating country voice.** Within the elevating country voice frame, the solution to the problem of suppressed country voice is improved LMIC representation and leadership in global health agenda setting (I5, I11, I14, I17, I27) [37]. This is reflected, for example, in the membership of Technical Working Groups and Steering Committees of global health initiatives implemented in LMICs—a strategy meant to elevate LMIC needs and priorities before programs are designed (I20). Referring to a new health initiative funded by a HIC organization, a respondent provided an example of improvement in representation as a solution to the suppressed voice of LMICs:

"*The steering committee is made up completely of colleagues from low- and middle-income countries, as opposed to colleagues from the global North or Westerners . . .*" (I11).

The elevating country voice frame also considers co-creation of health initiatives by funders and recipient countries as an indicator of progress (I10, I14) [9]. Another measure of elevated country voice is local actor's knowledge of local health programs, an indication of ownership (I16, I24). Country ownership is marked by engagement with national and subnational governments in identifying problems at the priority phase, and consultation in the designing of objectives and project activities. During the implementation phase, country ownership is marked by partnership in the implementation of actions, obtaining feedback, and

accountability through consultation in monitoring and evaluation. For example, the Local Engagement Assessment Framework (LEAF) of Oxfam America, Save the Children, and the Overseas Development Institute considers whether local governments are responsible for managing the project resources and whether they contribute resources (human resources in particular) to the project as markers of ownership [9].

In support of the strategy of greater representation and co-creation, a respondent suggested having:

"... *an open dialogue about asking for their help in identifying entrenched problems, where we could come up, jointly, with some ideas of what might help them overcome those entrenched obstacles or gaps in their programming, and then we would look for the innovation that might be relevant for that programming*" (I5).

Other identified markers of elevated country voice include: local actors taking leadership and decision-making roles in planning and implementing health initiatives and being able to push back when programs from HIC organizations don't make sense for their country (I15, I16, I18, I20). In addition, elevating country voice frames the use of national/sub-national government funding for programs (I11) [50] and the proportion of external funding that goes directly to local institutions as progress markers (I1, I14, I22).

The elevating country voice frame also proffers the adoption of locally funded initiatives—or putting funds directly in the hands of local actors—as a solution to the problem of the 'dollar power' (I3). A respondent explained:

"... *we are requiring that a minimum of 45% of funding that goes to our international partners or United States-based partners on MOMENTUM, that 45% of those funds are then sub granted to local partners on the ground*" (I14).

Another strategy involves improving the capacity of LMIC actors to take leadership in designing and implementing programs. Specifically, this includes clearly articulating an investment case for where LMICs want donors to provide support that can be more directly accessed by country actoea (I15); and redefining performance indicators to de-emphasize the interests of donors and focus on the priorities of LMICs (I15, I22). One respondent noted:

"*I think we have to redefine what we mean by 'results'*" (I22).

**Decolonizing global health.** The decolonizing global health prognosis is greatly nuanced ranging from calls to reform 'the system' to calls for 'disposing of it all together' [51]. A strand proffers–as a solution to the problem of racialized hierarchization of humanity and health systems–indigenous research with its roots in the worldview of indigenous people on an equal footing with Western scientific methods [52–54]. The indigenous methodology is known for equal power sharing between the researcher and the people studied, better community engagement, and respect for cultural beliefs, values, norms and practices. To address the problem of hierarchization of health systems, alternative home-grown solutions and alternative healing practices are to be sought and integrated into health systems to complement existing biomedical solutions [26, 40, 55]. Indigenous ways of knowing helps to address the problem of lack of equity in global health research [56]. The politics of knowledge acquisition needs to be transformed in order to address existing power asymmetry using several means, including encouraging that LMIC studies be published in local outlets or open access to keep knowledge within the reach of local actors (I15) and constantly keeping historical contexts in view. In the words of a respondent:

*"You have to just really understand the history of why people are where they are"* (I22).

The prognostic frame of decolonizing global health also considers the revision of medical and global health curriculums and program designs as essential to promote awareness of global health history, and to question and address negative traditional narratives, power asymmetry and white supremacist notions in global health [57, 58]. In addition, curriculums need to better emphasize patient safety, fair trade principles, diversity, equity and local and global contexts [54, 59].

The decolonizing global health frame also sees the need to address the problem of power asymmetry in global health governance. It recommends a structural transformation of global health governance that gives power to LMICs to fully control their own health systems (I18; I23; I26). One respondent suggested:

*"... it's time that we remove this colonial element out of aid, and we bring the power, as they say, back to where it belongs, in countries of the South"* (I23).

One respondent, stressing the need for LMICs to control their own programs, said:

*"... can we tell them to bugger off if they don't want to do what we want them to do or what we need them to do, right?"* (I26)

In the same vein, another respondent calls for a:

*"complete shifting of power to those whose lives need to be changed"* (I22).

By implication, the prognostic framing of decolonizing global health advocates for improved public health financing in LMICs (I1; I17). This view rests on the assumption that reliance on funds from HIC actors buys 'donors' the influence they need to control public policies and programs in LMICs [36]. A key proponent of decolonization considers that, in the near future, the solution may be to "*abolish global health*" in order for all actors to think of a new reality that is currently unknown (I26). The frame also sees the co-existence of biomedicine and alternative medicines within a health system as a marker of progress. At the global level, it sees the absence of racial and supremacist ideas in global health governance and initiatives as important markers of progress. More specifically, this will mean diversity, equity and inclusion in the boards of global health initiatives and major global health institutions.

The decolonizing global health diagnostic frame understands global health coloniality to be a part of a broader system characterized by power asymmetry. A broad-based approach that seeks to decolonize all facets of life is thus required for an impactful change to occur in global health (I25). Based on evidence that poverty is a driver of poor health outcomes, such an approach will also need to address the policies that continue to limit economic growth and keep populations impoverished in LMICs [49]. The decolonization of the minds of key LMIC actors is an essential component of this strategy, as suggested by a respondent:

*"the decolonizing of the mind' is another solution'* (I15).

The decolonizing global health prognostic frame will therefore see power symmetry in global health governance and a non-racist and non-hierarchized (i.e., egalitarian) world to be requirements for decolonizing global health.

### Motivational framing—Rationale for action

We present here how the frames underlying each of these concepts elaborate their call-to-action or their rationale for action.

**Elevating country voice.**   The motivational framing of elevating country voice is detailed in the 2005 Paris Declaration on Aid Effectiveness and similar documents that guide development practice involving many multilateral and bilateral organizations and LMICs that receive development aid [8, 9]. The call-to-action rests on the argument that aid to LMICs has the greatest impact when development programs align with local priorities and are owned and led by local actors. The elevating country voice frame calls actors to a gradual change in their approach to global health practice in LMICs. This call, if implemented, will result in a phased transition from global health practice where priorities and agenda are set by external funders to one in which local actors determine the priorities, own and implement local health initiatives, and contribute significantly to the resources used in the implementation of programs. The motivational frame also prioritizes the principles of harmonization of development work to avoid duplication, focusing on measurable results with specific performance indicators, and mutual accountability [8]. To achieve its goals, the motivational framing of elevating country voice calls for the strengthening of local capacity for exercising leadership in development programming, building more effective and inclusive partnerships, and achieving development results and openly accounting for them [17]. This motivational frame targets HIC institutions involved in funding programs in LMICs, local partners, and governments of recipient nations.

The motivational frame is summed up in the message:

> "*it's really hard to get things done without that [elevating country voice]*" (I20).

It is framed as a way of supporting LMIC institutions in what they do (I2) and as the only way to ensure sustainability (I18; I22). The framing is perceived by some as instrumental, presenting elevating country voice as the means to improving effectiveness of development work in LMICs.

**Decolonizing global health.**   A part of the motivational frame of decolonizing global health can be summarized in the demand that global health work be done "*as an act of justice and not charity or saviorism*" (I26). The motivational framing of decolonizing global health has been strongly expressed in commentaries and original articles in journals and blogs [3, 4, 37]. This frame typically emphasizes the 'injustice' and 'inequity' of global health practice, calling for an overthrow of the worldview on which global health practice currently rests. It presents this worldview as supportive of the notion of the universality of Western knowledge rather than a pluriversal worldview of humanity [27]. By implication, this call-to-action considers power and voice asymmetry in global health as symptoms of racialized hierarchization of humanity and health systems and exploitative liberalism.

However, because of the choice of message outlet, it often leaves out key actors involved in the implementation of global health initiatives in LMICs, as well as key government actors (I3, I10, I16) and sometimes does not resonate with LMIC actors (I26). Several respondents noted that the decolonization of health dialogue largely emerged and has been advanced in academic circles (I22, I27), especially from "*developing countries in academic institutions in high income countries*" (I27).

Calls for DGH may not resonate yet with many people from LMICs partly because of the mismatch in the preferred media outlets of the thought leaders and local actors, and partly because the current DGH discourse—which is narrower in reach than the broader decolonization discourse—is perceived to be heavily entrenched in HIC networks. One respondent noted how the ideas are yet to be acceptable to local actors:

*". . . we had colleagues from the Africa team say they're not comfortable using the 'decolonizing' concept"* (I27).

An LMIC respondent explained that local actors may have other priorities. They argued that:

*". . .if you came flinging 'decolonizing health systems', people might probably want to say, look we are more interested in maybe weeding out corruption, [achieving] some transparency, and knowing what the money is being used for"* (I16).

Another LMIC respondent, when asked whether the concept of decolonization resonates, clarified:

*"No. Number one, I would not have bought that [discourse] because they [those advancing 'decolonizing global health'] are not on the ground. . .. I am one of those people that are very careful with people that are not in the kitchen and they talk a lot about cooking . . . For any movement to be effective, it must be home-grown"* (I19).

The foregoing shows a tension within the decolonizing global health frame. While it speaks to the themes of power asymmetry, justice and fairness in respecting the views of those most concerned by global health programs, it appears to have left those most concerned about the matter of decolonizing global health behind–those in LMICs. Although the key sponsors of the frame among the respondents in this study are largely from LMICs, they currently operate in HIC institutions and their views may not resonate with LMIC actors in LMICs. They appear to be suggesting what is best for LMICs when they themselves are outside those LMICs, an idea that is inconsistent with the spirit of decolonization.

In summary, the motivational frame of decolonizing global health appears yet to be fleshed out, perhaps on purpose, in order to accommodate multiple views and create a larger coalition of actors who care about different facets of inequality and injustice.

## Discussion

The calls for elevating country voice and decolonizing global health both seek to achieve change by addressing power and voice asymmetry in global health practice. Our frame analysis shows that the former became expedient to address the problem of ineffective aid in addressing development problems in LMICs, while the lattersees the way aid is provided by HIC actors and the nature of representation in global health programs (which the elevating country voice advocates approve of as having failed) as part of the problem. The call for elevating country voice has clear progress markers that donors and HIC institutions involved in global health use as benchmarks. In contrast, the decolonizing global health frame does not appear to have clear indicators. Some of the proponents see aid to poor countries, or the conditions attached to the provision of assistance, as problematic, yet the frame is not strong enough in arguing for its discontinuation. Decolonizing global health has been framed as an act of justice, because the HICs are seen as responsible for the poor state of things in LMICs. However, unlike the elevating country voice frame, the decolonizing global health frame tends to downplay the role of poor LMIC public investment in public health initiatives in keeping local health systems dependent on foreign aid and in maintaining the hierarchy of medicines in former colonies.

The dissonance between the expectation that global health improves equity in healthcare access and the reality that it carries its predecessors' baggage of hierarchization, domination and exploitation is the reason behind the call for decolonization [6]. In line with the Paris

Declaration, global health initiatives are expected to be guided by a principle requiring donors to respect the leadership, ownership and expertise of local actors. By implication, LMICs should 'have a voice' in their own development programs. The growing call for global health decolonization suggests that making global health practice equitable may require more than 'elevating country voice'.

Our analysis shows that elevating country voice in global health practice is conceptually different from decolonizing global health and the best efforts to elevate country voices in global health practice are very unlikely to decolonize global health. That these concepts run parallel to each other is partly a result of their different *raison d'etre*–one having evolved out of a concern to make aid have greater impact, and the other, to address injustice and the absence of equity in global health practice.

The principles of the Paris Declaration and similar documents have continued to serve as a compass for global health practice among many global health actors. With the USAID Forward reform agenda it can be arugued that country voice elevation has begun. On the contrary, frame alignment, which is a requirement for a successful social movement [14, 15], is yet to occur with decolonizing global health. As our analysis shows, the idea does not sufficiently resonate with many key actors, especially in LMICs. The implication is that the decolonizing global health movement may still be far from achieving the change it recommends as there is little evidence that LMIC governments and other local actors are convinced of the need to decolonize. The study does not show that the decolonizing global health movement is attempting to or succeeding with convincing LMIC governments to spend more on health as one of the requirements for decolonization.

We argue that the decolonizing global health frame does not have a sufficiently clear prognostic or motivational frame. We can infer that its proposed solution is wholesale change in the global social, political and economic order that privileges HICs and disadvantages LMICs as opposed to an incremental change. At present, the enormity of this type of change leaves many actors confused on how to address the problem. In addition, the decolonizing global health movement's prognosis remains open to multiple interpretations with varied implications for global health practice, with proponents emphasizing different aspects. If, for instance, it is interpreted to mean stopping aid from HICs only, in the face of the daunting challenge of bringing about wholesale change in the social, political and economic order simultaneously, the question that follows naturally is whether LMIC public investment in health will be increased sufficiently to fill the vacuum that will be created by this withdrawal of aid.

## Conclusion

Views clash in the elevating country voice versus decolonizing global health discourse. Among actors interviewed in this study, origin and nature of involvement with global health work are strong markers of their diagnostic, prognostic and motivational framing of global health practice. HIC actors in global health practice often embrace the elevating country voice frame. A major implication of this is that the structure and mode of operation of major organizations involved in health interventions in LMICs have begun to change in line with the set standards in the Paris Declaration, although it remains unknown whether this is yielding the desired result of elevated country voice. The LMIC actors interviewed welcome the idea of elevated country voice. A major weakness of the decolonizing global health frame, on the other hand, is its prognostic frame and the preferred channel for disseminating its motivational message. Another important question in the discourse has to do with whether a wholesale, broad-based decolonization of all facets of life that is required to achieve an impactful change in global health is feasible. This call appears to offer an 'all-or-nothing' solution to power asymmetry in

global health. Even if the proposed wholesale change occurs, it may not translate to LMIC national governments' commitment to improved public health investment. Yet, the global health decolonization discourse hardly emphasizes this matter of health investment by governments in LMICs and its role in achieving full decolonization of global health. This question remains yet unanswered, and the answer holds much for the future of global health governance and practice.

## Supporting information

**S1 File. Standards for reporting qualitative research checklist.**
(DOCX)

## Acknowledgments

US Agency for International Development (USAID) and the study participants.

## Author Contributions

**Conceptualization:** Michael Kunnuji, Yusra Ribhi Shawar, Rachel Neill, Malvikha Manoj, Jeremy Shiffman.

**Formal analysis:** Michael Kunnuji, Malvikha Manoj.

**Funding acquisition:** Yusra Ribhi Shawar, Jeremy Shiffman.

**Investigation:** Michael Kunnuji, Yusra Ribhi Shawar, Rachel Neill, Malvikha Manoj, Jeremy Shiffman.

**Methodology:** Michael Kunnuji, Yusra Ribhi Shawar, Rachel Neill, Malvikha Manoj, Jeremy Shiffman.

**Project administration:** Michael Kunnuji, Yusra Ribhi Shawar, Rachel Neill, Jeremy Shiffman.

**Resources:** Michael Kunnuji, Jeremy Shiffman.

**Software:** Michael Kunnuji.

**Supervision:** Yusra Ribhi Shawar, Jeremy Shiffman.

**Validation:** Jeremy Shiffman.

**Writing – original draft:** Michael Kunnuji.

**Writing – review & editing:** Michael Kunnuji, Yusra Ribhi Shawar, Rachel Neill, Malvikha Manoj, Jeremy Shiffman.

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
