## [Decision Letter · Decision Letter 0]

24 Oct 2022

PGPH-D-22-01120

Why ‘elevating country voice’ is not decolonizing global health: a frame analysis of in-depth interviews

Dear Dr Michael Kunnuji

Thank you for submitting your manuscript to PLOS Global Public Health. After careful consideration, we feel that it has merit but does not fully meet PLOS Global Public Health’s publication criteria as it currently stands. Therefore, we invite you to submit a revised version of the manuscript that addresses the points raised during the review process.

We look forward to receiving your revised manuscript.

Kind regards,

Andreas K Demetriades, MBBChir, MPhil, FRCSEd, FEBNS.

Academic Editor

Journal Requirements:

1. Please provide your detailed Financial Disclosure statement. This is published with the article. It must therefore be completed in full sentences and contain the exact wording you wish to be published.

a. Please clarify all sources of funding (financial or material support) for your study. List the grants (with grant number) or organizations (with url) that supported your study, including funding received from your institution. 

b. State the initials, alongside each funding source, of each author to receive each grant.

c. State what role the funders took in the study. If the funders had no role in your study, please state: “The funders had no role in study design, data collection and analysis, decision to publish, or preparation of the manuscript.”

d. If any authors received a salary from any of your funders, please state which authors and which funders.

2. Your manuscript is missing the following sections: Abstract. Please ensure these are present, and in the correct order, and that any references to subheadings in your main text are correct. An outline of the required sections can be consulted in our submission guidelines here: [LINK]

https://journals.plos.org/globalpublichealth/s/submission-guidelines#loc-parts-of-a-submission

4. In the online submission form you indicate that your data is not available for proprietary reasons and have provided a contact point for accessing this data. Please note that your current contact point is a co-author on this manuscript. According to our Data Policy, the contact point must not be an author on the manuscript and must be a third party. Please revise your data statement to a non-author institutional point of contact, such as a data access or ethics committee, and send this to us via return email. Please also include contact information for the third party organization, and please include the full citation of where the data can be found.

Additional Editor Comments (if provided):

A very encouraging peer review.

For minor revisions please.

Reviewers' comments:

Reviewer's Responses to Questions

**Comments to the Author**

1. Does this manuscript meet PLOS Global Public Health’s publication criteria? Is the manuscript technically sound, and do the data support the conclusions? The manuscript must describe methodologically and ethically rigorous research with conclusions that are appropriately drawn based on the data presented.

Reviewer #1: Yes

Reviewer #2: Yes

2. Has the statistical analysis been performed appropriately and rigorously?

Reviewer #1: N/A

Reviewer #2: N/A

3. Have the authors made all data underlying the findings in their manuscript fully available (please refer to the Data Availability Statement at the start of the manuscript PDF file)?

Reviewer #1: Yes

Reviewer #2: Yes

4. Is the manuscript presented in an intelligible fashion and written in standard English?

Reviewer #1: Yes

Reviewer #2: Yes

5. Review Comments to the Author

Reviewer #1: The article includes an excellent review of relevant literature and seamlessly links it with key stakeholder interviews. The article is well written and contributes to our understanding of the frames of lifting country voice and decolonization. The nuanced distinction that is drawn would be of interest to global health practitioners and students.

My only suggestion/question: Is it possible to give a geographic distribution of interviewees without compromising anonymity? For example in the methods could you say that xx/28 were in the Americas, xx/28 in Asia, etc? That would present an additional level of nuance that would be of interest to readers.

Reviewer #2: The authors did an extensive work and this proves the scientific relevance of the paper. As a final comment, I would suggest the authors to add in the text the definition of “country voice” as used in the document. To a reader without First hand knowledge might not be accustomed with the framework and who the "Country voice" represent.

R

6. PLOS authors have the option to publish the peer review history of their article (what does this mean?). If published, this will include your full peer review and any attached files.

**Do you want your identity to be public for this peer review?** For information about this choice, including consent withdrawal, please see our Privacy Policy.

Reviewer #1: **Yes: **Meghan Farley Webb

Reviewer #2: No

---

## [Decision Letter · Decision Letter 1]

23 Jan 2023

Why ‘elevating country voice’ is not decolonizing global health: a frame analysis of in-depth interviews

PGPH-D-22-01120R1

Dear authors

We are pleased to inform you that your manuscript 'Why ‘elevating country voice’ is not decolonizing global health: a frame analysis of in-depth interviews' has been provisionally accepted for publication in PLOS Global Public Health.

Best regards,

Andreas K Demetriades, MBBChir, MPhil, FRCSEd, FEBNS.

Academic Editor

Thanks for your resubmission which peer review has found suitable for publication

Reviewer Comments (if any, and for reference):

Reviewer's Responses to Questions

**Comments to the Author**

1. If the authors have adequately addressed your comments raised in a previous round of review and you feel that this manuscript is now acceptable for publication, you may indicate that here to bypass the “Comments to the Author” section, enter your conflict of interest statement in the “Confidential to Editor” section, and submit your "Accept" recommendation.

Reviewer #1: All comments have been addressed

Reviewer #2: All comments have been addressed

2. Does this manuscript meet PLOS Global Public Health’s publication criteria? Is the manuscript technically sound, and do the data support the conclusions? The manuscript must describe methodologically and ethically rigorous research with conclusions that are appropriately drawn based on the data presented.

Reviewer #1: Yes

Reviewer #2: Yes

3. Has the statistical analysis been performed appropriately and rigorously?

Reviewer #1: Yes

Reviewer #2: N/A

4. Have the authors made all data underlying the findings in their manuscript fully available (please refer to the Data Availability Statement at the start of the manuscript PDF file)?

Reviewer #1: Yes

Reviewer #2: Yes

5. Is the manuscript presented in an intelligible fashion and written in standard English?

Reviewer #1: Yes

Reviewer #2: Yes

6. Review Comments to the Author

Reviewer #1: The previous version of the article was very well-written. The minor edits the authors have made have only improved the article. I caught only one typographical error.

Line one of Intro has a typo- a normative (not anormative)

Reviewer #2: Based on my assessment I am satisfied that the authors have addressed all issues highlighted in the previous review

7. PLOS authors have the option to publish the peer review history of their article (what does this mean?). If published, this will include your full peer review and any attached files.

**Do you want your identity to be public for this peer review?** For information about this choice, including consent withdrawal, please see our Privacy Policy.

Reviewer #1: **Yes: **Meghan Farley Webb

Reviewer #2: **Yes: **Action Amos
